# Serine peptidase Vpr forms enzymatically active fibrils outside *Bacillus* bacteria revealed by cryo-EM

Yijia Cheng[1], Jianting Han[1], Meinai Song[1], Shuqin Zhang [1] & Qin Cao [1] ✉

Bacteria develop a variety of extracellular fibrous structures crucial for their survival, such as flagella and pili. In this study, we use cryo-EM to identify protein fibrils surrounding lab-cultured *Bacillus amyloiquefaciens* and discover an unreported fibril species in addition to the flagellar fibrils. These previously unknown fibrils are composed of Vpr, an extracellular serine peptidase. We find that Vpr assembles into fibrils in an enzymatically active form, potentially representing a strategy of enriching Vpr activities around bacterial cells. Vpr fibrils are also observed under other culture conditions and around other *Bacillus* bacteria, such as *Bacillus subtilis*, which may suggest a general mechanism across all *Bacillus* bacterial groups. Taken together, our study reveals fibrils outside the bacterial cell and sheds light on the physiological role of these extracellular fibrils.

Proteins can assemble into fibril forms that play important physiological or pathological roles, such as cytoskeleton[1], amyloid fibrils[2], and cytoophidia[3]. Likewise, outside of bacterial cells, various types of protein fibrils with essential functions exist, including flagella[4], pili[5], and functional amyloid[6,7]. Bacteria can adhere to material surfaces and form biofilms by embedding themselves into extracellular matrices composed of proteins, polysaccharides, DNA, and extracellular fibrils[8]. While the structure and function of many extracellular fibrils have been extensively studied[9–12], it remains unclear whether there are undiscovered fibrils outside of bacteria cells. Cryoelectron microscopy (cryo-EM) with helical reconstruction has been shown to be a powerful tool for determining near-atomic resolution structures of protein fibrils, and when applied to native biological samples, it can also identify previously undiscovered fibrils at the molecular level[13,14].

In this study, we extracted extracellular fibrils from lab-cultured *Bacillus amyloiquefaciens*, a Gram-positive bacterium with great potential in biotechnological applications such as plant growth promotion, biocontrol, and commercial enzyme production[15]. Using cryo-EM, we determined the structures of these fibrils and discovered a previously unreported fibril species composed of a serine peptidase Vpr. Our findings demonstrate the power of cryo-EM in identifying structures and functions of protein fibrils, and it may have

important implications in *Bacillus*-related biotechnological applications.

## Results

### Fibril extraction from *Bacillus amyloiquefaciens* biofilm for cryo-EM study

To extract fibrils for cryo-EM structure determination, *Bacillus amyloiquefaciens* grown on agar plates were collected (Supplementary Fig. 1a). From the cryo-EM micrographs, we observed two fibril species with distinct fibril morphologies, both ~15 nm in width (Fig. 1a, blue and green arrows), We also observed fibrils approximately 8 nm in width, but these fibrils were all bundled together and unsuitable for cryo-EM structure determination (Supplementary Fig. 2a, gray arrows). The two-dimensional (2D) classes could be categorized into three groups, with two of them representing two 15-nm-wide fibril species that were further identified as flagella and Vpr fibrils by atomic model building (Supplementary Fig. 2b, see next subsection). The third group of 2D classes represented fibrils that cannot be identified via cryo-EM, including the bundled 8-nm-wide fibrils described above. The distribution of these three groups in 2D classification is shown in Fig. 1. The cryo-EM map resolution for both two fibril species was 2.9 Å (Supplementary Fig. 2c, d), and data collection and processing statistics were listed in Table 1.

[1]Bio-X Institutes, Key Laboratory for the Genetics of Developmental and Neuropsychiatric Disorders, Ministry of Education, Shanghai Jiao Tong University, Shanghai 200030, China. ✉e-mail: caoqin@sjtu.edu.cn

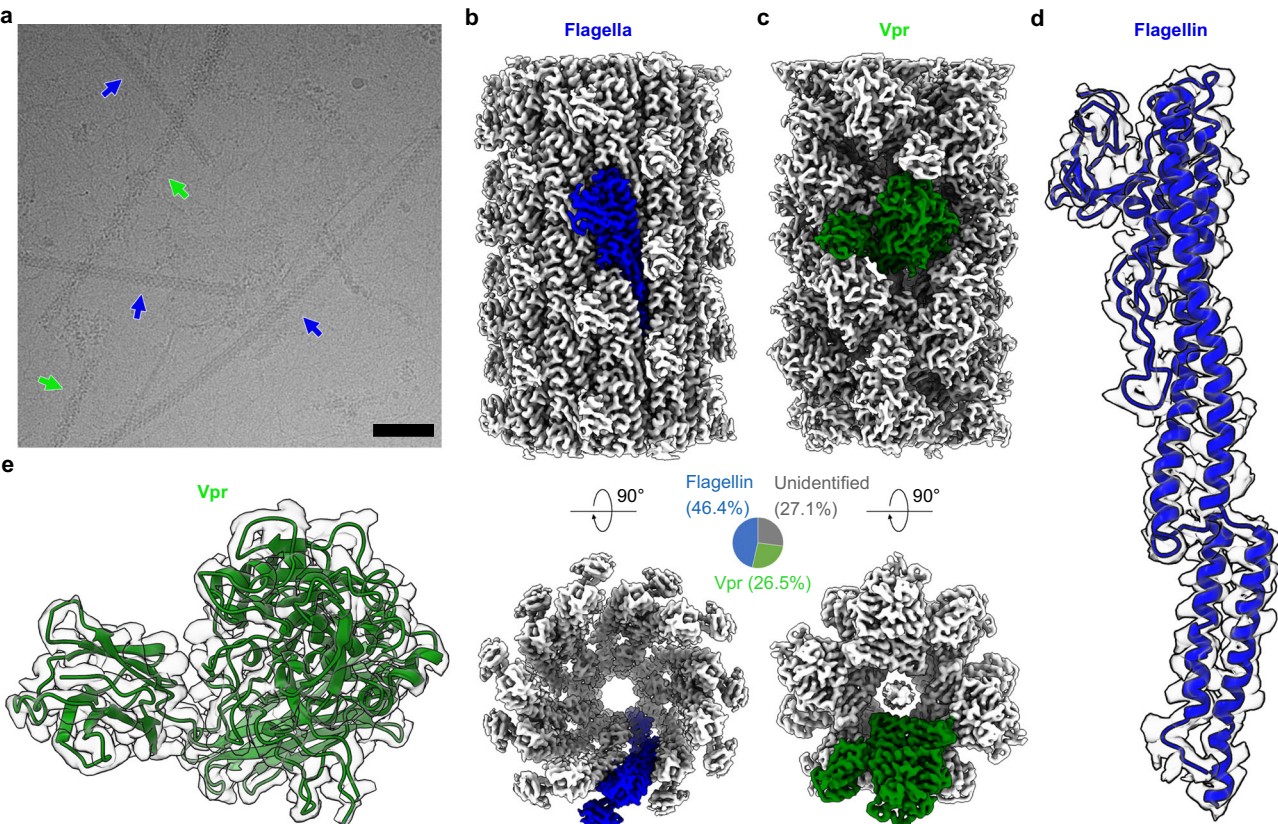

**Fig. 1 | Cryo-EM structures of flagella and Vpr fibrils. a** Representative cryo-EM micrograph collected in this study, indicating representative fibrils categorized as flagella (blue) and Vpr (green) with arrows. The scale bar is 100 nm. A total of 2894 micrographs has been collected in this study, and flagella (blue), Vpr (green) fibrils can be observed in most of these micrographs. **b, c** The cryo-EM maps of flagella (**b**) and Vpr (**c**) with each map from a single asymmetric unit colored. Pie chart showing the distribution of the two fibril species (in addition to the unidentified fibrils) was shown as an insert. **d, e** The cryo-EM maps and atomic models of one subunit from flagella (**d**) and Vpr (**e**), respectively.

## Atomic model building and identification of the fibril proteins

We used the strategy described in Supplementary Fig. 1 and Methods to build the atomic models and identify the fibril proteins for these two fibril species. The first fibril species was identified to be flagellar fibrils due to the similarities of its 2D classes and 3D map to that of *Bacillus subtilis* flagella[16]. For the other fibril species, because its 2D classes and 3D maps were not similar to previously reported ones, we first built the model using ModelAngelo, a graph neural network-based software capable of automatically building atomic models from cryo-EM maps without prior knowledge of protein identity[17]. The amino acid sequence from the ModelAngelo model was then used to search for proteins with similar sequences, and *Bacillus amyloiquefaciens* Vpr, a extracellular S8 family serine peptidase, was the top hit (sequence identity 77%). The final models were built with flagellin and Vpr sequences obtained from genotyping (Fig. 1b–e and Supplementary Fig. 1). The protein identities were further supported by the following observations: (i) for both two fibril species, the final models fit the cryo-EM maps well, leaving no unexplained density (Fig. 1d, e and Supplementary Fig. 3a, b), except for two extra densities that may suggest unknown ligand bindings or post-translational modifications of Vpr fibrils (Supplementary Fig. 3c); (ii) for Vpr fibrils, the ModelAngelo model sequences were used to search for proteins from all organisms, and Vpr from *Bacillus amyloiquefaciens* were the top hits; (iii) the final model for Vpr contains a well folded catalytic triad typical for serine protease (Fig. 2a, b). In summary, we identified two protein fibrils from *Bacillus amyloiquefaciens* biofilm using cryo-EM structure determination, which are composed of flagellin and Vpr, respectively.

## Structure of *Bacillus amyloiquefaciens* flagellar fibrils

The arrangement of subunits in *Bacillus amyloiquefaciens* flagella is similar to that of flagella from other bacteria, such as *Bacillus subtilis*[16] (Supplementary Fig. 4a). Upon structural comparison, we observed differences between *Bacillus amyloiquefaciens* and *Bacillus subtilis* flagella, namely: (1) the former contains an additional D2 domain, whereas the later contains only D0 and D1 domains; (2) the later has three additional loops (L1, L2, and L3) compared to the former (Supplementary Fig. 4b). These differences arise from the sequence variations between the two proteins (sequence identity 57%, Supplementary Fig. 4c). Notably, these variable regions face outward from the flagellar fibrils and do not affect the assembly of flagella (Supplementary Fig. 4b, d). In summary, our findings suggest that while the structures of flagella in *Bacillus amyloiquefaciens* and *Bacillus subtilis* are largely conservative, there are notable differences in their domain compositions that likely reflect sequence variations between the two proteins.

## Structure of Vpr fibrils

Vpr proteins assemble into fibrils with a helical symmetry and a $C_2$ symmetry. The helical symmetry contains left-handed 1-start, right-handed 4-start, and left-handed 6-start helices (Supplementary Fig. 5a). The fibrils are held together by two sets of inter-subunit interactions, one between subunit n and n+1 and the other between subunit n and n-2′ (the $C_2$ symmetry counterpart of n-2, Fig. 2b). These inter-subunit interactions are predominantly hydrophilic, including hydrogen bonds and salt bridges, but there are also

**Table 1 | Cryo-EM data collection, refinement and validation statistics of bacterial fibrils**

| | Flagella (EMD-36427, PDB 8JMV) | Vpr (EMD-36428, PDB 8JMW) |
|---|---|---|
| **Data collection and processing** | | |
| Magnification | ×130,000 | ×130,000 |
| Voltage (kV) | 300 | 300 |
| Electron exposure (e⁻/Å²) | 40 | 40 |
| Defocus range (μm) | 0.6–4.6 | 0.6–4.6 |
| Pixel size (Å) | 1.05 | 1.05 |
| Symmetry imposed | $C_1$ | $C_2$ |
| Helical rise (Å) | 4.8 | 20.5 |
| Helical twist (°) | 65.4 | −68.8 |
| Initial particle images (no.) | 318,365 | 318,365 |
| Final particle images (no.) | 153,285 | 76,671 |
| Map resolution (Å) | 2.9 | 2.9 |
| FSC threshold | 0.143 | 0.143 |
| Map resolution range (Å) | 200-2.9 | 200-2.9 |
| **Refinement** | | |
| Initial model used (PDB code) | 5WJT | De novo |
| Model resolution (Å) | 3.1 | 3.1 |
| FSC threshold | 0.5 | 0.5 |
| Model resolution range (Å) | 200-3.1 | 200-3.1 |
| Map sharpening $B$ factor (Å²) | 69 | 34 |
| Model composition | | |
| Non-hydrogen atoms | 800,025 | 69,552 |
| Protein residues | 10,692 | 9,306 |
| Ligands | 0 | 0 |
| $B$ factors (Å²) | | |
| Protein | 73.75 | 89.16 |
| Ligand | – | – |
| R.m.s. deviations | | |
| Bond lengths (Å) | 0.007 | 0.007 |
| Bond angles (°) | 1.034 | 0.640 |
| Validation | | |
| MolProbity score | 1.87 | 1.87 |
| Clashscore | 12.85 | 9.12 |
| Poor rotamers (%) | 0 | 0 |
| Ramachandran plot | | |
| Favored (%) | 96.25 | 94.35 |
| Allowed (%) | 3.75 | 5.65 |
| Disallowed (%) | 0 | 0 |

hydrophobic interactions (Fig. 2b and Supplementary Fig. 5d). Structural comparisons between the cryo-EM model and the alpha-fold model suggest that Vpr largely maintain the same conformation when assembled into fibrils, assuming that the alpha-fold model can represent the unassembled state of Vpr (Supplementary Fig. 5c). Each Vpr subunit contains residues 158-802 visible in cryo-EM model (Fig. 2a), which is consistent with the range of residues in the mature form of Vpr after proteolytic activation, according to previous studies of *Bacillus subtilis* Vpr[18] (residues 161-806 in *Bacillus subtilis* Vpr, corresponding to residues 158-803 in *Bacillus amyloiquefaciens* Vpr, Supplementary Fig. 6). Vpr proteins fold into four domains: a catalytic domain containing the typical catalytic triad of serine protease,

an insert domain visible only in the cryo-EM map with a low threshold, and two assembly domains mainly responsible for inter-subunit interactions (Fig. 2 and Supplementary Fig. 5b). There is an inner tunnel of ~23 Å in width at the center of Vpr fibrils, and extra densities are found inside the tunnel (Supplementary Fig. 5e). The shape of the extra densities is a straight tube with an approximately circular cross-section, and the diameter of the circular cross-section is ~15 Å (Supplementary Fig. 5e). These extra densities may suggest binding of unknown ligands, which should be negatively charged, as multiple lysine residues surround these densities (Supplementary Figs. 5e and 6, blue arrows). The relative position of the catalytic triad (Asp186, His230, and Ser531) suggests that the active site of Vpr protein is well folded and may be in a functional conformation. The active site in each subunit of Vpr fibrils is located in a cleft on the surface of the fibrils and is accessible to the solvent (Fig. 2b and Supplementary Fig. 5b). These observations suggest that Vpr may maintain its peptidase activity when assembled into fibrils.

**Vpr fibrils were enzymatically active**

To assess the activity of Vpr fibrils, we synthesized a 15-residue substrate peptide derived from PhrC, a precursor peptide reported to be the substrate of Vpr[19]. When incubated with fibrils extracted from *Bacillus amyloiquefaciens*, we detected the 5-residue proteolytic product, similar to previous reports[19] (Fig. 3a). We note that this enzyme activity was less likely to originate from the monomer form of Vpr or other extracellular peptidase such as subtilisin or Epr[19]. During fibril extraction, most soluble proteins should be removed by centrifugation and washing with a centrifugal concentrator (see Methods). Therefore, we believe that the Vpr fibrils we extracted are in an enzymatically active form, consistent with our structure analysis.

**Vpr fibrils existed under other culture conditions and in other *Bacillus* bacteria**

*Bacillus amyloiquefaciens* was grown under two additional culture conditions, and identical Vpr fibrils were observed under both conditions (Fig. 3b). Additionally, Vpr fibrils were found in *Bacillus subtilis* strain 168. These observations suggest that Vpr fibrilization may be a common feature among the *Bacillus* bacteria group. This hypothesis is further supported by sequence alignment, which shows that Vpr sequences are conservative within the *Bacillus* bacteria group (sequence identity above 84%), especially for residues involved in inter-subunit interactions (Supplementary Fig. 6, green arrows). It is worth noting that no extraction methods such as homogenization were used in these cryo-EM imaging experiments with *Bacillus* bacteria, and Vpr fibrils were found outside the bacteria cells, confirming their extracellular locations.

## Discussion

In this study, we extracted fibrils from lab-cultured *Bacillus amyloiquefaciens* and identified these fibrils using cryo-EM structure determination. Cryo-EM has proven to be a powerful tool for identifying undisclosed protein fibrils from biological samples[13,14], and in this study, we applied this tool to bacteria and identified a previously unreported fibril species (Vpr), in addition to flagellar fibrils that have been extensively studied before. Interestingly, we found that the morphologies of flagella and Vpr fibrils were only distinguishable under cryo-EM observation (Fig. 3b), but not negative stain EM (Fig. 3c). This is likely because both of fibril species were ~15 nm in width, and the width remained consistent along the fibril axis. This may explain why Vpr fibrils had not been discovered until this study, as they could be considered as flagella fibrils under negative stain EM observation (Fig. 3c). Our study not only provides another example of the advancement of using cryo-EM to discover biological macromolecules but also suggests that there may be many previously unknown fibrils in bacteria that could have important functions.

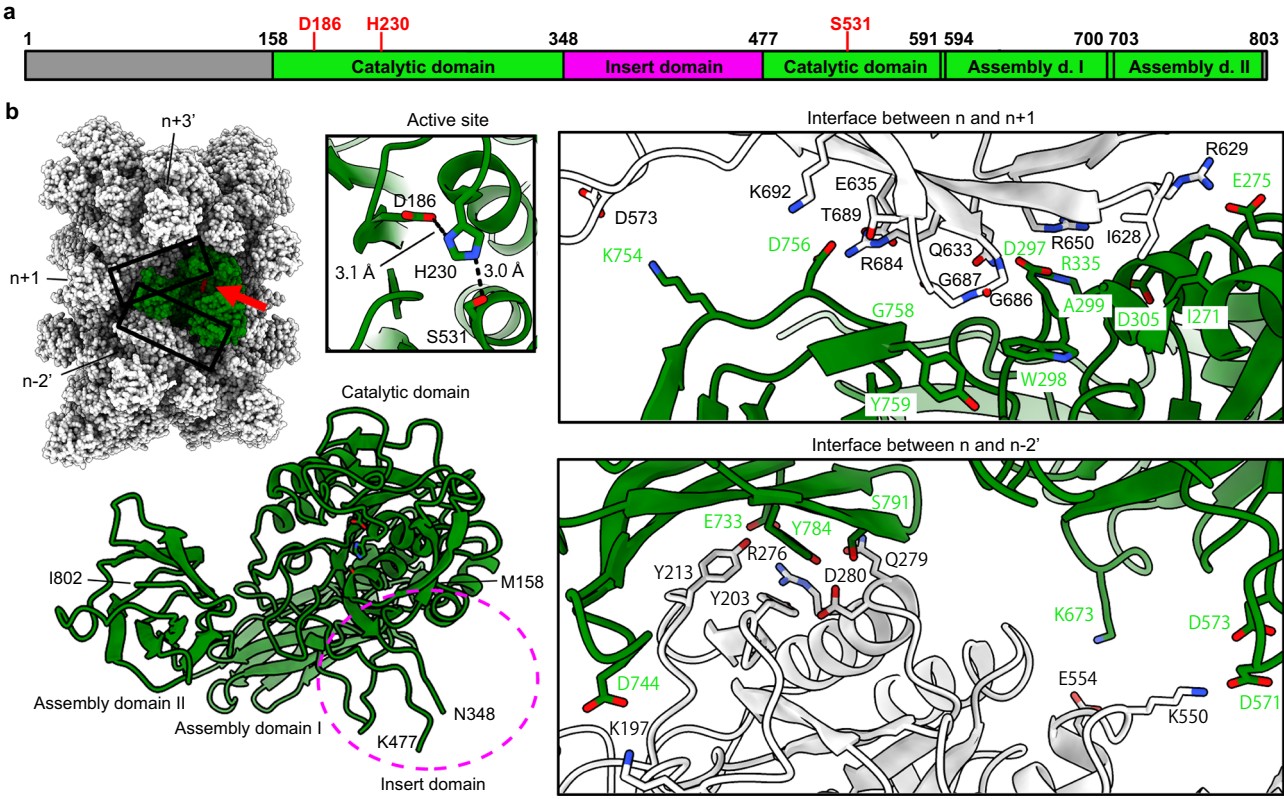

**Fig. 2 | Atomic model of Vpr fibrils. a** Schematic of Vpr with annotated residue ranges for the catalytic domain, insert domain, assembly domain (d.) I, and assembly domain (d.) II. The catalytic triad of Vpr is labeled in red. **b** Active sites and subunit interfaces of Vpr fibrils. The top-left panel shows the Van der Waals surface of the Vpr fibril structure, with subunit n colored in green and its active site colored in red. Adjacent subunits (n+1, n−2′, and n+3′, also see Supplementary Fig. 5a) are labeled. Two subunit interfaces are indicated with frames, and the active site of subunit n is indicated with a red arrow. The bottom-left panel shows the model of one subunit of Vpr in cartoon mode. The side chains of the catalytic triad are shown as sticks, and the magenta circle indicates the position of the insert domain. The right panels show the detailed structures of the active site and subunit interfaces, with the side chains of residues involved in inter-subunit interactions shown as sticks.

Vpr was reported to be an extracellular peptidase that cleaves precursor peptides to produce signal peptides (such as Competence and Sporulation Factor), which are involved in cell-cell communication and quorum sensing[19]. Bacteria use quorum sensing to regulate their physiological activities, including antibiotic production, motility, sporulation, and biofilm formation, critical for their survival[20]. In this study, we found that Vpr can form fibrils in *Bacillus amyloiquefaciens* and *Bacillus subtilis*, and perhaps in all *Bacillus* bacteria. Our structure shows that Vpr proteins maintain their native fold when assembled into fibrils, and their active sites are accessible in fibril form. These observations, together with our peptide cleavage assays, indicate that Vpr proteins retain their peptidase activity in fibrils and further suggest that these fibrils may have a functional role outside of bacteria cells. We propose that the function of Vpr fibrils is to concentrate and immobilize active Vpr proteins within extracellular matrices, thereby enhancing their ability to trigger quorum sensing. The initiation of quorum sensing depends on a minimal concentration threshold of signal molecules, which necessitates bacteria cells to be in close proximity to each other. When assembled into fibrils, Vpr activity can be easily enriched and extended over a long distance without dilution, as we observed Vpr fibrils with lengths up to serval micrometers. It is plausible to suggest that Vpr fibrils produced by one bacteria cell can extend several micrometers to neighboring cells and produce signal peptides, thereby triggering quorum sensing at a distance. Based on this analysis, we hypothesis that the function of Vpr fibrils is to facilitate long-distance cell-cell communication and lower the cell density threshold required for quorum sensing. In addition, secreted enzymes have been considered as a "public good" in bacterial community[21], and

a recent study suggested that extracellular proteases of *Bacillus subtilis* (including Vpr) are the public goods essential for bacteria growth under polymer nutrient source[22]. In this case, we propose that fibrilization of Vpr may enhance its ability as the public good and increase long-distance nutrient digestion. The importance of Vpr fibrils were further supported by the abundance of these fibrils found in *Bacillus amyloiquefaciens* biofilm, as the distribution of Vpr fibrils were comparable to that of flagellar fibrils (Fig. 1). In addition to quorum sensing, we note that Vpr fibrils may also have other physiological functions. For instance, a previous study suggests that Vpr has fibrinolytic activity[23], and another study suggests that Vpr can process TapA, which is involved in TasA fibril formation in *Bacillus subtilis* biofilm[24]. *Bacillus* bacteria are important for plant growth and have huge application potential for sustainable agriculture[25], and our findings may inspire future research to pay more attention to the functions of Vpr in this aspect.

Within the central cavity of the Vpr fibril, we observed tubular extra densities with an approximate diameter of 15 Å, suggesting ligand binding (Supplementary Fig. 5e). Structure analysis suggests that the bound ligands are likely to be negatively charged, and the diameter of the extra densities suggests that the ligand size is comparable to that of a continuous α-helix (with an average Van der Waals diameter of ~12–14 Å). However, due to the limited resolution of the extra densities, we cannot confirm the identity of the bound ligand in this study. Interestingly, the enzymatic activity of Vpr does not appear to be effected by ligand binding, as the active sites are located on the surface of Vpr fibrils, opposite to the position of the extra densities (Supplementary Fig. 5b, e). It remains uncertain whether ligand

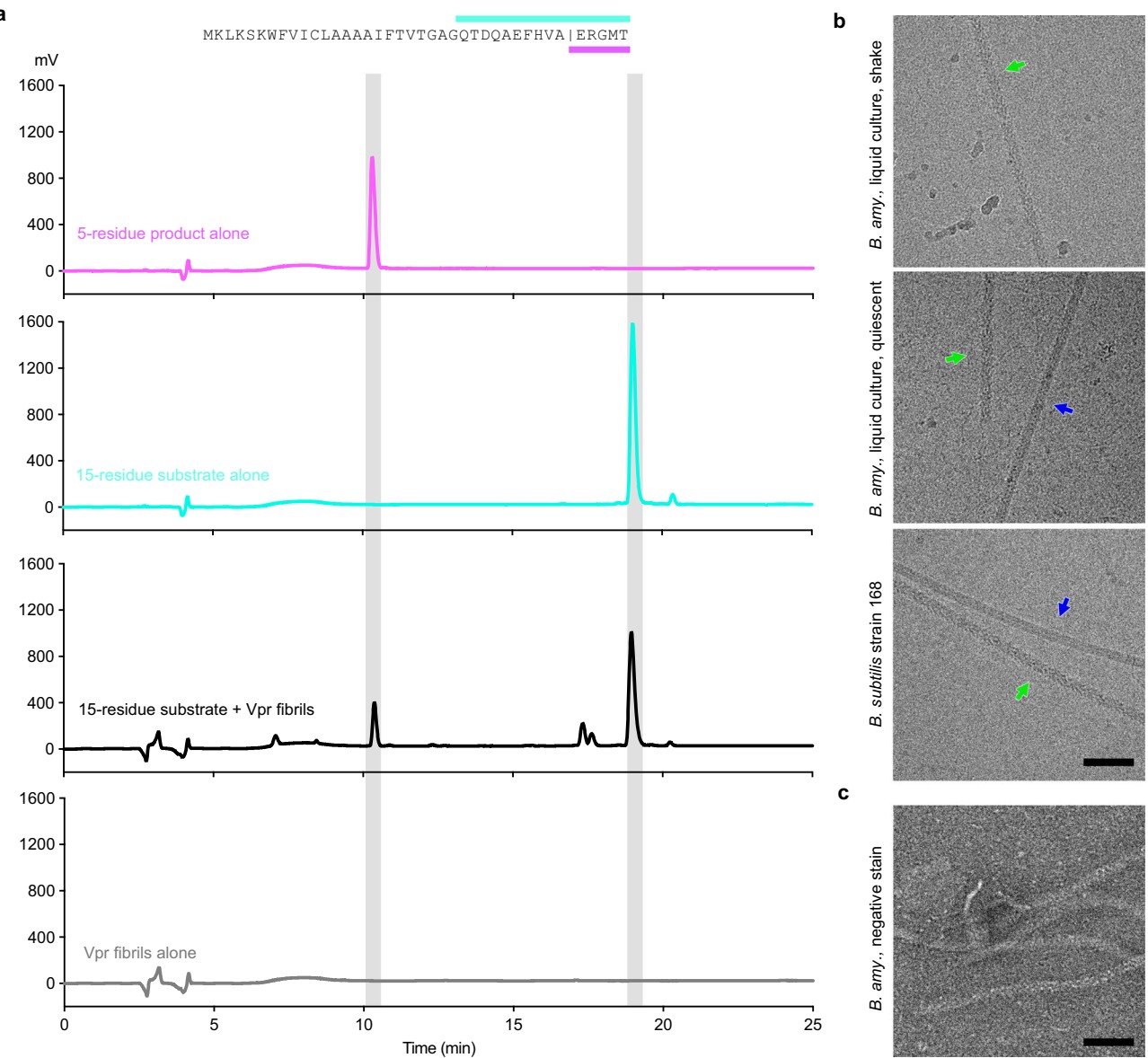

**Fig. 3 | Characterization of Vpr fibrils. a** Peptide cleavage assays of Vpr fibrils. The top panel shows the amino acid sequence of the PhrC peptide from *Bacillus amyloiquefaciens*. The cyan and purple lines on top and bottom of the sequence indicate the range of the 15-residue substrate and 5-residue product peptide, respectively, derived from *Bacillus amyloiquefaciens* PhrC. The vertical line in the PhrC sequence indicates the cleavage site of Vpr peptidase. The bottom panels show the HPLC spectra of the indicated samples. When the 15-residue substrate peptide was incubated with Vpr fibrils, a peak corresponding to synthesized 5-residue product peptide was observed on the HPLC spectra, suggesting that the cleavage occurred during incubation. The gray shades are time indicator markers. mV, millivolt. **b** The cryo-EM micrographs of fibrils from indicated bacteria and culture conditions. The green arrows indicate Vpr fibrils, and the blue ones indicate flagellar fibrils. **c** A representative negative stain micrograph of *Bacillus amyloiquefaciens* fibrils. We note that under negative stain EM, it is hard to distinguish flagella and Vpr fibrils. The scale bar is 100 nm. Each micrograph in panel b and c is a representative of at least three similar micrographs collected from the same sample. *B. amy.*, *Bacillus amyloiquefaciens*; *B. subtilis*, *Bacillus subtilis*.

binding has an impact on Vpr fibril formation and further studies are needed to identify the ligand and investigate its roles in the physiological functions of Vpr fibrils.

In summary, our study identified two protein fibrils outside *Bacillus* bacteria: the extensively studied flagella and the previously undiscovered Vpr fibrils. Our findings revealed that Vpr proteins assemble into fibrils with an enzymatically active conformation, indicating a unique physiological role of Vpr fibrils. Our findings provide insights into the diverse functions of bacterial fibrils in extracellular matrices, which may have implications for developing biotechnology applications related to *Bacillus* bacteria.

## Methods

### Bacteria used in this study

*Bacillus amyloiquefaciens* were gifts from Prof. Hao Yang's lab at Lanzhou University, China. *Bacillus subtilis* strain 168 was purchased from Hangzhou Baosai Biotechnology Co. (China, catalog number Y003). *Bacillus amyloiquefaciens* was initially identified as an unknown species from the genus *Bacillus*, and 16S rRNA sequencing failed to identify its species due to the highly conserved nature of the *Bacillus* gene[26]. To determine the species or strain of our bacteria, we performed sequence alignments using 16 S rRNA and genes encoding flagellin and Vpr that obtained from genotyping. We found four

*Bacillus* strains that have whole genome sequences available online and have identical sequences of 16S rRNA and genes of these two proteins, namely *Bacillus velezensis* JJ-D34, *Bacillus velezensis* YA215, *Bacillus amyloiquefaciens* GXU-1, and *Bacillus siamensis* IMD4001. These results suggest that the bacteria we used here should be or at least close to either of these four strains. *Bacillus velezensis*, *Bacillus amyloiquefaciens*, and *Bacillus siamensis* are phylogenetically closely clustered and can be referred to as the "operational group *Bacillus amyloiquefaciens*"[27]. Pan-genomic studies were required to distinguish these three *Bacillus* species and identify the strain of our bacteria, whereas we believe it is not necessary in this study because these three *Bacillus* species have very similar physiological features and our findings were not limited to a specific bacterial species or strain (Vpr fibrils were found also in *Bacillus subtilis* strain 168). Therefore, here we refer to the bacteria we used here as *Bacillus amyloiquefaciens* based on the name of their operational group, but we note that we do not have the species or strain information for these bacteria at the current stage. Meanwhile, *Bacillus subtilis* strain 168 is a standard bacteria strain, so we did not perform additional experiments to identify its strain.

## Bacteria culture

To culture *Bacillus amyloiquefaciens* for fibril extraction, a frozen stock was streaked onto an Luria–Bertani (LB) agar plate and incubated overnight at 37 °C. A single colony was then inoculated into LB broth and cultured overnight at 37 °C. The culture was diluted to an optical density ($OD_{600}$) of 0.01 and shaken at 37 °C for 30 min. Next, 80 μL of the diluted culture was spotted onto YESCA agar plates (1 g/L yeast extract, 10 g/L casamino acids, 20 g/L agar) and incubated at 25 °C for 3 days.

To investigate whether Vpr fibrils were also produced by other *Bacillus* bacteria, such as *Bacillus subtilis*, *Bacillus subtilis* strain 168 was cultured in YESCA agar plates using the same protocol. To investigate whether Vpr fibrils were produced under different culture conditions, *Bacillus amyloiquefaciens* was grown in LB broth at 37 °C with shaking for 3 days, and in Msgg broth (100 mM Mops, pH 7, 0.5% glycerol, 0.5% glutamate, 5 mM potassium phosphate, 50 mg/L tryptophan, 50 mg/L phenylalanine, 2 mM $MgCl_2$, 700 μM $CaCl_2$, 50 μM $FeCl_3$, 50 μM $MnCl_2$, 2 μM thiamine, and 1 μM $ZnCl_2$) at 37 °C without shaking for 3 days, respectively.

## Fibril extraction

To extract flagella and Vpr fibrils from *Bacillus amyloiquefaciens*, bacteria were scraped off from the YESCA agar plates and mixed with homogenization buffer (200 mM NaCl, 10 mM Tris-HCl, pH 7.4) before being homogenized on ice using a handheld homogenizer for 10 min. Sample solution was then centrifuged at 3700×*g* at room temperature for 10 min, and the pellet was discarded. The resulting supernatant was referred to as the crude extract and stored at 4 °C for further use. To further purify the extracted fibrils, the crude extract was centrifuged at 15,000×*g* for 10 min, and the supernatant was concentrated using a 100 kDa cut-off centrifugal concentrator (Millipore, catalog number UFC910024) and washed 3 times with buffer containing 10 mM Tris-HCl, pH 7.4. The concentrated sample was centrifuged at 3700×*g* at room temperature for 5 min, and the supernatant was further centrifuged at 15,000×*g* at room temperature for 5 min. The pellet was resuspended in buffer containing 400 mM NaCl, 10 mM Tris-HCl, pH 7.4, and used for cryo-EM data collection.

## Negative stain transmission electron microscopy

Glow-discharged 200 mesh carbon coated copper grids (Beijing Zhongjingkeyi Technology Co., Ltd., catalog number BZ11022a) were used for transmission electron microscopy. In all, 2.6 μl of sample solution was applied to the grids and incubated for 2 min, then excess sample solution was blotted off using filter paper. Grids were then stained with 3.3 μl of 2% (w/v) uranyl acetate for 1 min and further washed with an additional 3.3 μl of 2% (w/v) uranyl acetate. The grids were air-dried for 2 min and imaged using a Talos L120C G2 transmission electron microscope (Thermo Fisher Scientific).

## Cryo-EM data collection and processing

Cryo-EM grids were prepared by applying 2.6 μl of sample solution onto glow-discharged Quantifoil 1.2/1.3 200 mesh electron microscope grids (catalog number N1-C14nCu20-01). The grids were plunge frozen into liquid ethane using a Vitrobot Mark IV (Thermo Fisher Scientific). Cryo-EM data were collected on a Titan Krios transmission electron microscope (Thermo Fisher Scientific) equipped with a K2 Direct Detection Camera (BioQuantum), operated with 300 kV acceleration voltage and energy filter of 20 eV. Super-resolution movies were collected with a nominal physical pixel size of 1.05 Å/pixel (0.525 Å/pixel in super-resolution movie frames) and a dose per frame of ~1.25 e-/Å². A total of 32 frames were taken for each movie, resulting in a final dose of ~40 e-/Å² per image. Automatic data collection was performed using SerialEM v.3.8.6 software.

Cryo-EM data processing workflow is shown in Fig. 1b. Motion correction and contrast transfer function estimation were performed using MotionCorr2[28] and CTFFIND-4.1.827[29], respectively. Particles were first manually picked for the first 200–300 micrographs and were then automatically picked using the topaz v.0.2.5 software package[30], with the results of the manual picking used as the training dataset. Helical reconstruction was performed with RELION 4.0[31]. Two sets of particles were extracted with box sizes of 686 and 320 pixels, respectively, and were further used for 2D classification with a tau_fudge value of 2. The inter-box distance for particle extraction was set to 10% of the box size.

From 2D classes, we found two distinct fibril species. The 2D classes of one species resembled those of *Bacillus subtilis* flagellar fibrils previously reported[9,16], so the reported helical symmetry was used for 3D reconstruction. For the other species, the helical symmetry was determined via visual inspection of the 2D classes.

For 3D refinement, particles from the selected 2D classes with a box size of 320 pixels were used for automatic high-resolution gold-standard refinement. CTF refinement and Bayesian polishing were performed, and the final reconstructions were generated by one or two rounds of additional golden-standard refinement. The map resolution was estimated using the 0.143 Fourier shell correlation (FSC) resolution cutoff, and data collection and processing statistics were listed in Table 1.

## Atomic model building

In this study, two near-atomic-resolution cryo-EM maps were generated. The first map was similar to previously reported ones, so a previously reported model (*Bacillus subtills* flagella N226Y[16], PDB ID 5WJT) was docked into this map as a rigid body. The second map was not similar to any previously reported ones, so de novo model was built with no protein identity information provided. The resolution of this maps was 2.9 Å, with clear densities of side chains, allowing for unambiguous de novo modeling. Automatic modeling was performed using ModelAngelo 1.0[17], with no protein sequence provided, resulting in a model that fits the cryo-EM map well for most side chains, except for some negatively charged residues (Supplementary Fig. 3), likely due to radiation damage.

The amino acid sequence of ModelAngelo-generated model was used to search for protein hits across all organisms, and the S8 family serine peptidase Vpr from *Bacillus amyloiquefaciens* was the top hit for this de novo model, with sequence identities of 77%. The protein identities of flagella and Vpr fibrils were confirmed as described above, and the amino acid sequences of these two fibril proteins were further confirmed by genotyping since the bacteria used were not a standard

strain. The models were built using COOT v.0.9.8.2[32] based on the initial models (either docked or de novo generated) and confirmed sequence (also see Supplementary Fig. 1) and were further refined with phenix.real_space_refine[33]. The final model was validated using MolProbity[34]. Both phenix.real_space_refine and MolProbity are incorporated in software package Phenix v.1.20.1-4887.

## Genotyping

A single colony of *Bacillus amyloliquefaciens* was picked from an LB plate and suspended in 200 µL of water. Primers were designed to amplify *hag* and *vpr* genes, which encode Flagellin and Vpr, respectively. PCR amplification was performed using Phusion High-Fidelity DNA Polymerase (New England Biolabs), and the PCR products were sequenced by Shanghai Sangon Biotech Co. The 16 S rRNA was also amplified using the universal primers 27 F and 1492 R and sequenced. The primer sequences and genotyping results are listed in Supplementary Tables 1 and 2, and the genotyping sequences were used to build atomic models and identify bacteria species.

## Cryo-EM imaging of flagellar and Vpr fibrils

To investigate the existence of Vpr fibrils under different culture conditions, *Bacillus amyloliquefaciens* was cultured with liquid LB broth (shaken) and Msgg broth (quiescent) and directly applied to cryo-EM grids for sample preparation. Additionally, to investigate whether Vpr fibrils could be produced by other *Bacillus* bacteria, *Bacillus subtilis* strain 168 growing on YESCA agar plates was scrapped off and mixed with buffer containing 10 mM Tris-HCl, pH 7.4, and applied to cryo-EM grids.

## Peptide cleavage assays of Vpr fibrils

In a previous study, it was found that *Bacillus subtilis* Vpr can process Competence and Sporulation Factor (CSF), a 5-residue peptide, from its precursor, and one of the precursor peptide was PhrC[19]. Here, we synthesized a 15-resude peptide with a sequence derived from *Bacillus amyloliquefaciens* PhrC and used it as a substrate for Vpr cleavage assays. The expected product CSF was also synthesized and used as a positive control for cleavage. Peptides were synthesized by GL Biochem Co., Shanghai, and dissolved in water before use.

To perform cleavage assays, 2.5 mg/ml of the 15-residue substrate peptide was incubated with or without extracted Vpr fibrils at 37 °C for 2 h and analyzed with high-performance liquid chromatography (HPLC, Shimadzu Co., Japan). The synthesized 5-residue product peptide (2.5 mg/ml) and extracted Vpr fibrils were also analyzed by HPLC. Samples were filtered through a 0.1 µm pore size filter (Jetbiofil Co., Guangzhou, catalog number FPV103013), and 10 µL of each sample was injected into a reverse-phase C18 column (Welch) and eluted with a gradient of 0–99% acetonitrile in water, supplied with 0.1% formic acid.

## Reporting summary

Further information on research design is available in the Nature Portfolio Reporting Summary linked to this article.

## Data availability

Cryo-EM map and atomic model present in this study have been deposited into the Worldwide Protein Data Bank (wwPDB) and the Electron Microscopy Data Band (EMDB) with accession codes PDB 8JMV and EMD-36427 for *Bacillus amyloliquefaciens* flagella, and PDB 8JMW and EMD-36428 for *Bacillus amyloliquefaciens* Vpr fibrils. The initial model used for atomic model building of *Bacillus amyloliquefaciens* flagella is available in Worldwide Protein Data Bank (wwPDB) with accession code 5WJT. Source data of HPLC spectra shown in Fig. 3a are provided as a Source Data file. Any other relevant data are available from the corresponding author upon request. Source data are provided with this paper.

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

## Acknowledgements

We thank H. Yang from Lanzhou University for sharing the *Bacillus amyloiquefaciens* bacteria. This work was supported by STI2030-Major Projects 2022ZD0212500 to Q.C. and the National Natural Science Foundation (NSF) of China (grant nos. 32271276) to Q.C. The authors thank for cryo-EM data collection at the Instrument Analysis Center (IAC), Shanghai Jiao Tong University. The authors acknowledge the National Facility for Translational Medicine (Shanghai) for support.

## Author contributions

Y.C. cultured *Bacillus amyloiquefaciens* and extracted fibrils. Y.C., J.H., M.S., and S.Z. prepared cryo-EM grids and collected cryo-EM data. Q.C., Y.C., and J.H. processed cryo-EM data and built the atomic models. Y.C. performed genotyping of *Bacillus amyloiquefaciens* proteins and peptide cleavage assays of Vpr fibrils. All authors analyzed the results and wrote the manuscript. Q.C. supervised the project.

## Competing interests

The authors declare no competing interests.
