## [Peer Review File · Nature Communications]

Serine peptidase Vpr forms enzymatically active fibrils outside
Bacillus bacteria revealed by cryo-EMREVIEWER COMMENTS

Reviewer #1 (Remarks to the Author):

In this paper the authors identify and characterise active fibres formed by a serine protease called Vpr. The fibres are found in at least two *Bacillus* species and across different conditions. The findings are clearly presented and the concept of active protease fibres to allow collation of the activity is interesting and highly plausible. The work also shows how cryo-EM can be used in a "mining" approach to identify new external macromolecules.

Minor comments

Line 28 - language around defining the term "biofilm" is not right. The biofilm is the cells embedded in the matrix, not the matrix alone. Check the sentence construction please.

Line 28 - language around defining the term "biofilm" is not right. The biofilm is the cells embedded in the matrix, not the matrix alone. Check the sentence construction please.

Line 44- (and elsewhere) no italics for species names when the subheading is italics

Line 62 (and elsewhere)- small s for subtilis

Line 150 - interesting proposition and very reasonable - a schematic might be useful in your discussion or does space not allow this?

Line 173 - Add in that proteases are a known public good (including reference as this fits with your discussion and point being made)

Line 285- what is a "reasonable" request? This is an open statement that is not clear.

Line 330- is this an experimental parameter that might induce the fibres to form?

Line 345- and elsewhere add in (w/v) or (v/v)

Line 406- what are the primer sequences?

Line 413 and elsewhere - indicate 5' to 3' directionality

Line 515- remove data interpretation from methods

Figure 3- check the colour codes as the middle green/blue and pink colour peaks are hard to see behind the grey time indicator marker

Reviewer #2 (Remarks to the Author):

The authors found that Vpr, an extracellular serine peptidase, forms fibrils on the outside of lab-cultured *Bacillus amyloiquefaciens* bacteria and they determined the structure of these fibrils by cryo-EM. This is a very interesting and unexpected finding. The solved structure of the Vpr fibril is novel and has not been described before. The structure has a good resolution of 2.9 Å, which enables to see many interesting details, including the active site.

As also shown by the authors, the fibrils are enzymatically active. The authors discuss some possible reasons for why the fibrils form, which seem reasonable and provide a good foundation for further investigations.

The Vpr fibrils have a width that is very similar to the width of flagellar fibrils; a possible reason for why the presence of these fibrils might have been overlooked previously. The authors also reconstructed the flagellar fibrils and found that they are different from *Bacillus subtilis* flagellar fibrils. However, the functional consequences of this difference is not clear.

There is a tubular extra density in the central cavity of the Vpr fibril, with a diameter of 23 Å. The authors speculate that it arises likely from negatively charged ligands (the inner cavity is lined with many lysines). This extra density does not seem to be involved in enzymatic activity as the substrate binding site and active site is on the surface, exposed to the solvent.

It would be helpful to mention how strong this extra density is compared to the rest of the density. Can the authors estimate the density, volume or mass per Vpr protein, to get an idea what the size of this ligand is? This is difficult to judge from Fig. S5e alone.

Overall, this is a very interesting study. The manuscript is well written and results are clearly presented.

We thank the reviewers for the helpful suggestions of our manuscript, which has significantly contributed to improving the quality of our work. We have revised our manuscript accordingly, including the following changes:

1. We have added the size analysis of the extra densities as requested by reviewer #2, including revised Fig. S5e, the fourth paragraph of the Results section, and an additional paragraph in the Discussion section.
2. We have revised Fig. 3 according to reviewers' comments.
3. We have added a brief discussion about "public good" in the second paragraph of the Discussion section under the suggestion of reviewer #1 (also two references were added).
4. Other minor revisions according to reviewers' suggestions.

All changes are highlighted in revised manuscript, and our point-by-point responses to the reviewers follow below:

Reviewer #1 (Remarks to the Author):

In this paper the authors identify and characterise active fibres formed by a serine protease called Vpr. The fibres are found in at least two *Bacillus* species and across different conditions. The findings are clearly presented and the concept of active protease fibres to allow collation of the activity is interesting and highly plausible. The work also shows how cryo-EM can be used in a "mining" approach to identify new external macromolecules.

Response: We thank the reviewer for the positive comments, and we agree with the reviewer that this work shows the potential of "mining" new macromolecules via cryo-EM, as we are currently trying to apply this tool to more biological samples. We again appreciate the reviewer's encouragement regarding our work.

Minor comments

Line 28 - language around defining the term "biofilm" is not right. The biofilm is the cells embedded in the matrix, not the matrix alone. Check the sentence construction please.

Response: We thank the reviewer for clarifying the definition of biofilm, and the corresponding sentence has been revised to "Bacteria can adhere to material surfaces and form biofilms by embedding themselves into extracellular matrices composed of proteins, polysaccharides, DNA, and extracellular fibrils."

Line 44- (and elsewhere) no italics for species names when the subheading is italics

Response: We have revised the corresponding subheadings accordingly.

Line 62 (and elsewhere)- small s for subtilis

Response: We are sorry for the mistakes made, and they have been corrected in the revised manuscript.

Line 150 - interesting proposition and very reasonable - a schematic might be useful in your discussion or does space not allow this?

Response: We thank the reviewer for agreeing that our proposition is interesting and reasonable. We are sorry but we are unsure about what kind of schematic the reviewer is suggesting. We hope our revised Figure 3 could readily be able to support our hypothesis here. We have separated the original Fig. 3b into two panels (Fig. 3b and Fig. 3c), so that it should better emphasize the differences between cryo-EM and negative stain EM, particularly in terms of differentiating flagella from Vpr fibrils. In the other words, the revised Fig. 3b&c suggests that flagella and Vpr fibrils are distinguishable under cryo-EM (Fig. 3b), but not negative stain EM (Fig. 3c), so that Vpr fibrils may not be discovered via negative stain EM. Additionally, we have also added “(Fig. 3b)” after sentence “only distinguishable under cryo-EM observation”, and “(Fig. 3c)” after the sentence “...could be considered as flagella fibrils under negative stain EM observation”. We hope the reviewer agree that these revisions can make the discussion clearer similarly as the schematic the reviewer suggested.

Line 173 - Add in that proteases are a known public good (including reference as this fits with your discussion and point being made)

Response: We thank the reviewer for bringing the “public good” concept to our attention. We find it fits our discussions, as Vpr fibrilization may increase its ability in long-distance nutrient digestion as a public good. We have added two references and a brief discussion in revised manuscript, and we hope it can inspire the following studies to further investigate the physiological functions of Vpr fibrils.

Line 285- what is a "reasonable" request? This is an open statement that is not clear.

Response: We have removed the word “reasonable” from revised manuscript, since we believe most requests should be reasonable, and we will properly respond to all of them.

Line 330- is this an experimental parameter that might induce the fibres to form?

Response: We agree with the reviewer’s concern that fibril extraction steps may affect the existences of the fibrils, which is, certain fibrils may be induced to form during fibril extraction. However, we believe it is less likely to be the case in our study, because we have also tried to image *Bacillus amyloiquefaciens* and *Bacillus subtilis* cells that directed applied to cryo-EM grids without homogenizing (and for those cells grown in liquid broth, even without scrapping, please see “cryo-EM imaging of flagellar and Vpr fibrils” section in Methods). Flagellar and Vpr fibrils were observed under these conditions, suggesting that these fibrils are present before any extraction steps.

Line 345- and elsewhere add in (w/v) or (v/v)

Response: We have added “(w/v)” for all “2% uranyl acetate”.

Line 406- what are the primer sequences?

Response: We have added the primer sequences in Methods of revised manuscript. We note that we did not add 16S rRNA primers because they are universal primers.

Line 413 and elsewhere - indicate 5' to 3' directionality

Response: We have added “(from 5' to 3’)” before each sequence listed in Methods.

Line 515- remove data interpretation from methods

Response: We have removed this data interpretation from Methods in revised manuscript, and we note that it does not affect the clarity of the manuscript as the similar statement has already been stated in the last paragraph of Result section.

Figure 3- check the colour codes as the middle green/blue and pink colour peaks are hard to see behind the grey time indicator marker

Response: We have revised Figure 3 to make the peaks clearer, via adjusting the transparency and color of the grey time indicator maker.

Reviewer #2 (Remarks to the Author):

The authors found that Vpr, an extracellular serine peptidase, forms fibrils on the outside of lab-cultured *Bacillus amyloiquefaciens* bacteria and they determined the structure of these fibrils by cryo-EM. This is a very interesting and unexpected finding. The solved structure of the Vpr fibril is novel and has not been described before. The structure has a good resolution of 2.9 Å, which enables to see many interesting details, including the active site.

As also shown by the authors, the fibrils are enzymatically active. The authors discuss some possible reasons for why the fibrils form, which seem reasonable and provide an good foundation for further investigations.

The Vpr fibrils have a width that is very similar to the width of flagellar fibrils; a possible reason for why the presence of these fibrils might have been overlooked previously. The authors also reconstructed the flagellar fibrils and found that they are different from *Bacillus subtilis* flagellar fibrils. However, the functional consequences of this difference is not clear.

There is a tubular extra density in the central cavity of the Vpr fibril, with a diameter of 23 Å. The authors speculate that it arises likely from negatively charged ligands (the inner cavity is lined with many lysines). This extra density does not seem to be involved in enzymatic activity as the substrate binding site and active site is on the surface, exposed to the solvent.

It would be helpful to mention how strong this extra density is compared to the rest of the density. Can the authors estimate the density, volume or mass per Vpr protein, to get an idea what the size of this ligand is? This is difficult to judge from Fig. S5e alone.

Response: We are sorry that we did not label the size of the extra densities in the original manuscript. We have revised Fig. S5e and added the label. The shape of the extra densities is a straight tube with an approximately circular cross-section, and the diameter of the circular cross-section is around 15 Å. It is measured when we set the threshold of the density to the same as that of the green densities shown in Fig. S5b (under this threshold the densities cover most of the visible main chain and side chains of Vpr). Judged from this diameter, the size of this ligand should be slightly bigger than a continuous α -helix (the average Van der Waals diameter is ~12-14 Å) and smaller than a B-form DNA (the average Van der Waals diameter is ~32 Å). We have added the shape and size description of the extra densities in the

Result section of revised the manuscript, and we have added an paragraph in Discussion section to further discuss these extra densities.

Overall, this is a very interesting study. The manuscript is well written and results are clearly presented.

Response: We thank the reviewer for the positive feedbacks of our manuscript.

REVIEWERS' COMMENTS

Reviewer #1 (Remarks to the Author):

Thank you for the amendments.

Reviewer #2 (Remarks to the Author):

In this revised manuscript, the authors have addressed my original comments adequately. I support publication in this current version.

typo:

page 5, line 196: "...comparable to that of a continu*ous* α -helix..."

We thank again for the reviewers' helpful suggestions and positive comments, and our point-by-point responses to the reviewers follow below:

Reviewer #1 (Remarks to the Author):

Thank you for the amendments.

Response: We thank the reviewer for the time and efforts.

Reviewer #2 (Remarks to the Author):

In this revised manuscript, the authors have addressed my original comments adequately. I support publication in this current version.

typo:

page 5, line 196: "...comparable to that of a continu*ous* α -helix..."

Response: We are sorry for the typo, and we have corrected it in the revised manuscript (the revised word is highlighted).